# Effects of Preservation and Propagation Methodology on Microcosms Derived from the Oral Microbiome

**DOI:** 10.3390/microorganisms10112146

**Published:** 2022-10-29

**Authors:** Baoqing Zhou, Jen Mobberley, Kelly Shi, Irene A. Chen

**Affiliations:** 1Department of Chemistry and Biochemistry, University of California, Santa Barbara, CA 93106, USA; 2Department of Chemical and Biomolecular Engineering, Department of Chemistry and Biochemistry, University of California, Los Angeles, CA 90095, USA

**Keywords:** oral microbiome, microcosm, plaque, cryopreservation

## Abstract

The creation of oral microcosms with reproducible composition is important for developing model systems of the oral microbiome. However, oral microbiomes vary substantially across individuals. To derive a reproducible composition from inocula sourced from different individuals, we tested whether selective conditions from cold storage and culturing in defined media would generate a reproducible community composition despite individual variations. In this pilot study, we collected dental plaque scrapings from three individuals, inoculated media under anaerobic conditions, and characterized the bacterial community compositions after cold storage and subsequent propagation in liquid media. Harvested cultures were extracted and bacterial composition was determined by 16S rRNA gene amplicon sequencing and the mothur pipeline. Our results show that samples from two out of three individuals clustered into a specific compositional type (termed “attractor” here). In addition, the samples from the third individual could adopt this attractor compositional type after propagation in vitro, even though its original composition did not display this type. These results indicate that simple selective environments could help create reproducible microcosms despite variation among dental plaque samples sourced from different individuals. The findings illustrate important parameters to consider for creating reproducible microcosms from the human oral microbiome.

## 1. Background

The oral microbiome is a key factor in oral health. Research over the past three decades has shown that the oral microbiome harbors opportunistic pathogens as well as commensal organisms [1,2,3] and that certain organisms (e.g., members of the *Porphyromonas* and *Streptococcus* genera) are associated with diseased states of the oral cavity [3,4,5,6]. Furthermore, oral microbes such as *C. albicans* and *Pseudomonas* spp. are also associated with systemic diseases [7]. These studies suggest that certain salivary bacterial biomarkers may serve as effective disease pre-indicators and trackers [7,8,9]. Therefore, just as understanding the composition of the gastrointestinal microbiome has led to advances in novel treatments of obesity [10] and inflammatory bowel diseases [11,12,13,14,15], deeper understanding and modeling of the oral bacterial community is important for developing novel therapeutic approaches [16], particularly for diseases that are influenced by multiple pathogenic agents.

Older studies on the oral microbiome used conventional techniques, including confocal microscopy, fluorescence microscopy, isolation/cultivation of individual organisms, PCR, gel electrophoresis, and DNA–DNA hybridization. These techniques characterized many aspects of the oral microbiome, including its biofilm nature [17], spatial heterogeneity [18,19,20], partial composition [5,18,19,21,22,23], host-associated compositional variations [24], general colonization order [22], organismal roles in community dynamics [23], and the association of genera such as *Porphyromonas, Tanerella,* and *Prevotella* with diseased states [6]. The potential significance of generating an in vitro community that mimics the in vivo community was also recognized [20,25,26]. In more recent years, high-throughput sequencing (HTS), especially using the 16S rRNA gene marker, has enabled facile and in-depth studies of both the memberships and the abundances of the oral microbiome. HTS protocols [27,28,29,30,31,32,33] are now well-established for the identification and characterization of the microbiome [34,35]. The ability to efficiently and rapidly identify microorganisms has in turn led to methodological improvements for culturing, both for single-strain cultures of difficult oral organisms and for generating more biologically relevant in vitro microbiomes. Advances include supplementing with specific compounds or assistive strains [26], extending cultivation times, depleting environmental factors that contribute to the growth of undesired microbes [25], and dilution [36].

Several efforts have been directed toward creating experimental models (microcosms) of the oral microbiome in vitro. A fruitful approach uses host plaque to inoculate in vitro cultures [37,38], with supplementation by pig mucin used to mimic salivary mucin [38,39]. In addition, the practice of adding specific nutrients has been effective in promoting the proliferation of fastidious organisms [38,40,41]. Modification of the media has been shown to favor different bacterial compositions, such as healthy or disease-associated communities [42]. Other practices, such as using oral surface mimics [37,43,44,45,46,47], constructing devices that imitate salivary flow conditions [39,48], and creating biologically defined inocula composed of laboratory strains [37,39,40,43,45,48], have been adopted. Some models in recent years have emphasized nutrient composition and the formation of an artificial pellicle [38,49,50,51]. However, the reproducibility of microcosms seeded by microbiome samples, particularly after preservation (e.g., cold storage) and propagation of the inoculum, has not been characterized well. While preservation experiments have been extensively performed on isolated bacterial species [52,53], preservation of human-associated microbiomes has primarily focused on the gut microbiome. Research on preserving the gut microbiome indicated that storing samples at various temperatures with different preservatives does not significantly impact community structures [54,55,56], although the effect on a seeded microcosm was not studied. Cryopreservation, especially with glycerol and inulin, can help maintain the composition and functionality of artificial gut microbiota [57,58], and indeed freeze-dried fecal samples after long-term storage can still be used for transplantation treatment of *C. difficile* infections [59]. Intra- and inter-subject variability outweighs technical variability, including variations due to sequencing runs or long-term storage [60]. Unlike the gut microbiome, relatively little research has been performed on the effects of preserving native oral microbiomes, particularly on their ability to produce a reliable microcosm. 

Since selective conditions are expected to influence the community composition, we hypothesized that cold preservation and subsequent culture propagation of a dental plaque sample would yield a similar bacterial community despite individual differences in the initial plaque sample. Such a community might be used to seed a microcosm of reproducible composition despite variation across individual plaque samples. To probe the potential stability of such experimental models of oral microbiomes, we studied how preservation (refrigeration or freezing) and cultivation of dental plaque samples affect the bacterial community’s composition. We generated in vitro microcosms derived from three different subjects to investigate the reproducibility of a microcosm derived from preserved, complex bacterial communities (Figure 1). We examined the community composition shifts that occurred when the host plaque was transmigrated into an in vitro environment and the compositional shifts that occurred as a result of preservation and in vitro propagation.

## 2. Methods

### 2.1. Sample Collection and Culturing of Human Plaque Communities

Plaque was collected from three volunteer hosts under protocols 3-18-0189 and 3-19-0119 approved by the UCSB Human Subjects Committee. Subjects were age 18–80, and information on gender was not collected. The collection process was as follows. Supragingival plaque scrapes of molar teeth from three healthy adult hosts were collected with sterile metal curettes, after hosts had abstained from food, drink, and dental hygiene practices for 12–16 h. The sites of the scrapes were limited to the mucosal supragingival surfaces of three molar teeth (upper or lower). The plaque from each host was used to inoculate 6 mL of SHI media [38], which had been centrifuged at 8000× *g* for 5 min to remove particulate contaminants. Inoculated media were divided equally into three wells in a 24-well surface-modified plate (Corning 353847) with each well receiving 2.0 mL of liquid. Prior to receiving the inoculated media, wells were preconditioned with a pellicle layer. To prepare the pellicle, pooled healthy human saliva was purchased as frozen aliquots (BioIVT), defrosted on ice, and clarified by centrifuging at 6000× *g* for 3 min, mixing with 1X PBS, pH 8.0, and passing through 0.2-micron syringe filter. The pellicle was formed by adding 150 μL of filtered clarified saliva to each well, incubating the plate at 37 °C for 1 h, and then sterilizing with short-wave UV radiation (254 nm) for 1 h. Inoculated SHI media from the three different hosts were then pipetted into three separate plates. Three negative control wells were also prepared in a fourth plate, with a salivary pellicle layer, 2.0 mL of SHI media, and no host plaque.

Plates were incubated at 37 °C in a sealed vessel (Figure 1a) in an anaerobic atmosphere of 85% nitrogen, 10% hydrogen, and 5% carbon dioxide. Every 24 h, approximately 1.3 mL of spent media was pipetted from the top of each well without disturbing the sedimented cells. An amount of 1.5 mL of fresh SHI medium was pipetted into each well, followed by addition of 20 μL of 0.5% sucrose. Afterwards, plates were returned to the anaerobic atmosphere for continued incubation until harvesting at 72 h. These 72 h cultures were termed “initial cultures”. Two of the three inoculated wells from the initial cultures were used in the subsequent preservation experiment. Contents of the third well were pelleted at 16,000× *g* for 5 min, flash-frozen in liquid nitrogen, and stored at −80 °C for later analysis.

We investigated several methods of preserving the microbial communities derived from the initial cultures. Two wells from the control plate and two wells from each host plate were selected from the initial culture plates. The entire volume of a single well (approximately 1.75 mL) was divided into five volumes, each measuring approximately 0.35 mL. Sterilized glycerol was added to one aliquot to a final concentration of 20%, and this sample was flash-frozen in liquid nitrogen for later analysis (“initial culture pellet”). The remaining four aliquots were each subjected to one of the following four preservation conditions: 4 °C for 1 day, 4 °C for 3 days, cryopreservation with 20% glycerol at −80 °C for 1.5 weeks, and cryopreservation with 20% glycerol at −80 °C for 5.5 weeks. After preservation, 180 μL was saved for subsequent propagation experiments. The remaining portion (approximately 170 μL) was divided into two equal aliquots, flash-frozen in liquid nitrogen with 40% glycerol, and stored at −80 °C for later analysis (“preserved cultures”).

To assess the revival and growth of microbial communities, we used the reserved 180 μL from each preservation condition to inoculate 6.5 mL of sterilized SHI medium. The inoculated media were divided equally among three wells in surface-modified plates, having been conditioned by a pellicle layer as detailed above. Cultures derived from different hosts were kept separate from one another. Plates were incubated at 37 °C for 48 h under the aforementioned anaerobic atmospheric conditions, wells were treated with procedures identical to those in the initial cultures, with replenishment of spent/partially spent media and sucrose supplementation. Of the three wells of the resulting 48 h cultures, henceforth termed “propagated” cultures, one was pelleted at 16,000× *g* for 5 min and flash-frozen in liquid nitrogen as reserve (not used). The other two were mixed with glycerol to a final concentration of 20%, pelleted at 16,000× *g* for 5 min, flash-frozen in liquid nitrogen, and stored at −80 °C until further processing.

### 2.2. DNA Extraction

Genomic DNA from initial cultures, preserved cultures, propagated cultures, positive controls, and negative controls, was extracted with the DNeasy PowerSoil kit (Qiagen 12888, formerly MoBio PowerSoil Kit). Negative controls included no-inoculum controls incubated alongside initial cultures, preserved cultures, and propagated cultures, as well as sterile SHI medium and extraction controls (200 μL of 1X PBS). Positive controls consisted of 200 μL of ZymoBiomics microbial community standard (D6300). The entire collection of samples, including culture controls and the microbial community standard, was randomly divided into batches of 11. Each batch was processed with an extraction control of 200 μL sterile 1X PBS, making a total of 12 samples per extraction batch. During each extraction, frozen pellets were thawed on ice, resuspended in 1X PBS, and pelleted at 6000× *g* for 5 min. The resuspension and pelleting steps were repeated two more times, for a total of three washes. Then, pellets were resuspended in 200 μL of 1X PBS, and DNA was extracted according to the manufacturer’s instructions.

The concentration of extracted DNA was measured using a fluorometric kit (QuantIT PicoGreen; Invitrogen). Estimation of post-extraction bacterial biomass was performed using the SSoAdvanced Universal SYBR Green Supermix qPCR assay (Biorad 1725271) using gene-specific primers designed to amplify the V4 region from the 16S rRNA gene (Appendix A). Quantitation by qPCR showed that 80 out of 82 non-negative-control samples contained greater than 10,000 copies per μL and that all negative controls contained 17 to 2885 copies per μL. These data confirmed that inoculated samples contained sufficient bacterial biomass for sequencing. The low concentrations of 16S rRNA in the negative controls were consistent with expected low bacterial biomass. All samples were included for sequencing.

### 2.3. Amplicon Library Construction and Illumina MiSeq Sequencing

16S rRNA gene amplicon libraries were constructed following a dual-index sequencing protocol [61] using specific gene primers (515F-805R) [62,63,64]. Amplicons were generated on 96-well plates using 1 μL of template DNA, 1 μL of each index primer, and 17 μL of Accuprime Pfx Supermix. Each plate contained a negative control well (1 μL of molecular grade water) and a positive control well (1 μL of ZymoBiomics community DNA standard). PCR was performed under the following conditions: initial denaturation at 95 °C for 2 min, followed by 30 cycles of denaturation at 95 °C for 20 s, annealing at 55 °C for 15 s, extension at 72 °C for 1 min followed by a final extension at 72 °C for 10 min. Amplicon purification using AMPure XP beads was performed followed by normalization and pooling to equimolar amounts of each sample. Each sample was sequenced in duplicate on different plates. Illumina MiSeq sequencing with PE300 V3 chemistry was performed in the genetics core of the Biological Nanostructures Laboratory within the California NanoSystems Institute.

### 2.4. 16S rRNA Bioinformatics Analysis

The mothur software package (v. 1.40) was used to process the paired-end Illumina 16S rRNA gene reads [61,65]. Briefly, read pairs were assembled into contigs, and contigs with ambiguous reads were excluded such that there were 13,249,077 total contigs. The contigs were aligned to the Silva SSU reference non-redundant database (version 132) [66], and those that did not align were removed. A denoising algorithm [67] and chimera identification with UCHIME [68] were performed as additional quality control steps, resulting in 11,662,068 high quality sequences. The contigs were clustered at the 3% dissimilarity level to generate operational taxonomic units (OTUs). Sequences were classified against the Silva SSU reference non-redundant database (version 132) using a naive Bayesian classifier (80% pseudo-bootstrap confidence score cutoff). The ZymoBIOMICS Community DNA served as our “mock community” to calculate sequencing error rates. When these mock communities were examined using the above pipeline, we observed an overall error rate of 0.019277% and 8 OTUs (compared to 8 expected bacterial species in the mock community), indicating that the read processing pipeline has a low error rate and does not overestimate diversity. The mothur code (i.e., batch file) processed fasta files, and count tables used to generate the BIOM table are freely available as described in the Availability of Data section.

### 2.5. Statistical Analysis

All statistical analyses were performed with R (version 4.0.3) in RStudio (version 1.3.959). The phyloseq package (version. 1.34.0) was used in the initial analysis of the OTU data generated by mothur [69]. Before analyzing the cultures, we examined negative controls, including all culture controls and DNA extraction controls, for potential contamination. To do so, we compared the number of reads and OTUs between negative controls and cultures. We also compared actual read distributions of positive controls—commercial mock microbial and DNA standards—with their expected distributions. After verifying that results from negative and positive controls were within expectation, we examined sequencing depth by host and by sample type. To account for differences in sequencing depth, we also examined the number of OTUs by hosts and by sample types. We rarefied samples to an even depth of 240 reads per sample, the lowest number of reads among cultures, and observed the number of OTUs post rarefaction. To investigate the effects of rarefaction on the number of OTUs, we constructed rarefaction curves for all 82 culture samples, excluding one outlier with more than 700,000 reads and two samples with fewer than 1000 reads. To determine whether any correlation existed between sequencing depth and the number of OTUs, we plotted the number of reads against three diversity indices, omitting controls, mock communities, samples with fewer than 1000 reads, and an outlier with more than 700,000 reads. To determine the most common phyla across these same samples, we studied prevalence plots of sequencing depth vs. taxa prevalence.

After confirming that rarefaction was valid for the analysis, we examined sample compositions by plotting relative abundances according to preservation conditions for each host. Principal coordinate analysis (PCoA) was performed on the relative abundances using the Bray–Curtis dissimilarity metric [70]. A dot plot of relative abundances, averaged by preservation conditions and categorized by host, was used to inspect the five most prominent OTUs. Principal component analysis (PCA) was also performed on the relative abundances of initial, preserved, and propagated cultures, and the PCA scree plot was used to assess the contribution to total variation from components 3 and beyond. To assess the validity of performing PCA on relative abundance data, we transformed the data with the centered-log-ratio (CLR) and isometric log-ratio (ILR) operations from the version 6.18.0 of the mixOmics package [71]. Additional PCA was performed with these transformed data. We performed significance tests using the ANOSIM function [27,72] in the vegan package (version 2.5.7) [73], with Bray–Curtis distance measures and 999 permutations, to test whether and how much preservation conditions and/or host differences influenced the cross-sample differences in relative abundances.

## 3. Results

### 3.1. Sample Collection, Controls, and Rarefaction 

Samples were collected from three hosts following the scheme outlined in Figure 1. Plaque samples were used to inoculate SHI growth media [42] supplemented by a saliva-based pellicle. Cultures were grown for 72 h (“initial cultures”), and aliquots were removed and subjected to storage in one of four conditions: 4 °C for 1 day, 4 °C for 3 days, −80 °C with 20% glycerol for 1.5 weeks, and −80 °C with 20% glycerol for 5.5 weeks (“preserved cultures”). Aliquots were then taken for additional propagation after storage (“propagated cultures”).

As expected, negative culture controls (media and pellicle only) contained <1500 reads (Appendix A), with 28 out of 33 samples having <800 reads, confirming low read counts in negative controls compared to the cultures (Appendix A). The ZymoBIOMICS microbial community standard served as a positive control for both the extraction and sequencing processes, and the ZymoBIOMICS microbial community DNA standard served as an additional positive control for the sequencing process. Expected relative distributions of both standards, generated from the manufacturer’s specifications, are shown in Appendix A. 16s rRNA sequencing does not yield eukaryotic reads, so two eukaryotic organisms from the DNA standard are not graphically represented. Comparison of the theoretical distribution to experimental distribution of the DNA standard (Appendix A, gDNA standard) shows a reasonable match with all species detected. Observed distributions for the microbial standard (cells) deviated more from the theoretical distribution (Appendix A, microbial standard). These greater deviations were expected due to the additional variation in DNA extraction efficiency across different bacterial species (e.g., DNA being more efficiently extracted from Gram-negative species compared to Gram-positive species). Nevertheless, all species were detected in the positive control. In addition, the most prevalent phyla in the oral samples were Actinobacteria, Bacteroidetes, Firmicutes, Fusobacteria, and Proteobacteria (Appendix A), corresponding to phyla of the most commonly expected members of the oral microbiota, which confirms the validity of the experimental and bioinformatics procedures.

To check whether the read depth of a sample affected measurement of community diversity, we plotted three diversity indices (Shannon index, inverse Simpson’s index, and richness) against the number of reads per sample. No significant correlation was observed for the Shannon or inverse Simpson’s indices, but a statistically significant positive correlation (*R*^2^ = 0.07) was observed between read counts and richness (Appendix A). This correlation suggests that richness comparisons across samples with varying read counts may be affected by variations in sequencing depth, with higher read counts leading to apparently increased richness. We examined rarefaction curves for all samples (Appendix A) and found that the number of operational taxonomic units (OTUs) increased as the read counts increased. To correct for the OTU overestimation in high read count samples, we rarefied samples to an equal depth of 240 read counts (lowest read count in the samples) and checked the number of species-level OTUs before and after rarefaction (Appendix A). The number of OTUs decreased for most samples upon rarefaction. For the microbial community standard (Appendix A, Mock_ext), eight organisms were expected, and rarefaction removed spurious OTUs, as the OTUs identified after rarefaction matched those expected from the theoretical composition (Appendix A). Therefore, rarefaction reduced artifacts from variable sequencing depth and was kept in the subsequent bioinformatic analysis.

### 3.2. Decrease in Community Diversity from Initial Plaque Culturing 

Plaque samples from all three hosts included two OTUs from the *Actinomyces* genus and at least one OTU from each of the following taxa: *Corynebacterium*, *Rothia*, *Streptococcus*, and *Veillonella* (Figure 2). In terms of evenness of the distribution of relative abundances, Host 2 plaque differed substantially from Hosts 1 and 3. In Host 2, *Streptococcus* OTU001 clearly dominates, and *Fusobacterium* OTUs were absent. In Hosts 1 and 3, *Streptococcus* OTU005 was much more prominent than OTU001, and a *Fusobacterium* OTU was present. Host 1 and Host 3 plaque also contained distinctively higher abundances of the *Actinomyces*, *Corynebacterium*, and *Rothia* OTUs than Host 2 plaque, while the *Veillonella* OTU was more abundant in Host 2 than in Hosts 1 and 3. In addition, *Prevotella* OTUs appeared only in Host 1 and the uncultured F0332 genus appeared only in Host 3 plaque.

Anaerobic incubation of plaque-inoculated SHI media for 72 h generated the initial cultures. In these cultures, the diversity of the community decreased in all hosts compared to their respective plaque inoculum. Of the 11 taxa present in host plaque, only three to five taxa appeared in the initial cultures. Members from the taxa of *Actinomyces*, *Corynebacterium*, F0332, *Fusobacterium*, and *Rothia* disappeared, yielding to members in the *Prevotella, Streptococcus,* and *Veillonella* genera (Figure 2, Cx Pellet). The composition of Host 1 initial cultures diverged from those of Hosts 2 and 3, as *Veillonella* OTU002 dominated the Host 1 initial cultures while *Streptococcus* OTU001 dominated the initial cultures of Host 2 and Host 3. Furthermore, the two duplicates of Host 1 differed from each other, as *Prevotella* OTU006 increased in one duplicate of Host 1 (Pellet 2), while an increase in an *Alloscardovia* OTU was observed in the other (Pellet 1). For all hosts, the initial culturing step represented a significant selection step, yielding a small subset of taxa from the plaque inocula, although the subset was not yet consistent across hosts or duplicates.

### 3.3. Effect of Cold Preservation on Community Composition 

When the initial cultures were subjected to four different preservation conditions (−80 °C × 1.5 weeks, −80 °C × 5.5 weeks, 4 °C × 1 day, 4 °C × 3 days), dominant members of the community for all three hosts largely remained dominant regardless of preservation conditions (Figure 2, “Pres” samples). However, preservation did change the relative abundances of these members in some cases. Preserved cultures of Host 1 still contained four major OTUs, including two *Streptococcus* and one *Veillonella* taxa that were present in both initial culture wells and two other OTUs specific to individual wells (*Alloscardovia* OTU011 for Pellet 1 and *Prevotella* OTU006 for Pellet 2); Host 2 and Host 3 contained two *Streptococcus* OTUs and one *Veillonella* OTU, as the initial cultures did. In all three hosts, the highest abundance taxon remained dominant (*Streptococcus* OTU001 in Hosts 2–3 and *Veillonella* OTU002 in Host 1), regardless of preservation conditions. The effect of cold preservation on *Alloscardovia* OTU011 (Pellet 1) and *Prevotella* OTU006 (Pellet 2) in Host 1 was of special interest since these species had been found in only a single initial culture across hosts, raising a possibility that these may not be stable members of the community. *Alloscardovia* OTU011 (Pellet 1, Host 1) was observed after preservation at −80 °C but was substantially decreased in abundance after preservation at 4 °C. However, *Prevotella* OTU006 (Pellet 2, Host 1) survived across all preservation conditions. 

### 3.4. Propagation of Cold-Preserved Cultures Favors an “Attractor” Community Composition across Individual Samples

When cold-preserved cultures were revived and propagated in fresh media, relative abundances changed in systematic ways (Figure 2, “Prop” samples). In all hosts, propagation appeared to favor *Streptococcus* OTU001. Propagation also appeared to favor *Alloscardovia* OTU011 (Host 1, Pellet 1). Indeed, a few of the cultures propagated from Pellet 2 of Host 1, whose initial culture did not contain a detectable amount of *Alloscardovia* OTU011, yielded this OTU after propagation. In contrast, in Host 1, Pellet 2, *Prevotella* OTU006 was essentially eliminated after propagation in most samples. Exceptionally, one sample (P2, −80 °C × 1.5 weeks) showed increased *Prevotella* OTU006, displacing the typically dominant *Streptococcus* OTU001. This divergence among samples suggested that there may be more than one steady-state composition after propagation. 

In Hosts 2 and 3, *Streptococcus* OTU001 became unequivocally dominant in the propagated cultures, and in some cases, it became the only visibly present OTU. Both *Streptococcus* OTU005 and *Veillonella* OTU002 often decreased in relative abundance. Overall, propagation of the preserved cultures tended to further reduce diversity, with a shift toward communities including the *Streptococcus*, *Veillonella*, and occasionally *Prevotella* taxa.

Principal coordinate analysis (PCoA) with Bray–Curtis dissimilarity distances [70] was used to visualize similarities and differences in community compositions (Figure 3). The first two principal coordinates accounted for more than 97% of the total variation in the samples. Interestingly, all cultured, preserved, and propagated samples from Host 2 and Host 3 formed a cluster at the same location, although Host 2 samples clustered less tightly than Host 3 samples. In stark contrast, samples from Host 1 formed multiple clusters (including one overlapping the cluster of Hosts 2/3). Of the Host 1 clusters, two clusters—one containing some of the propagated cultures and the other containing the initial and preserved cultures—showed distinct compositions compared to Hosts 2/3. The third cluster in Host 1, containing most of the propagated cultures, overlapped the region occupied by Hosts 2/3. This indicates that propagation led to a shift in community composition toward the Hosts 2/3 state. This composition thus appears to act as an “attractor” state as an initially diverse community undergoes in vitro culturing, preservation, and propagation. Indeed, while Host 2 showed some dispersion in compositions compared to Host 3, the propagated samples of Host 2 clustered more tightly at this “attractor” state (PC1 at approximately −0.2) compared to the other samples. Samples from Host 1 showed that initial cultures, and cold preservation without propagation, did not produce communities with the “attractor” composition. Differences in preservation conditions did not clearly translate into compositional differences. In addition, for Host 1, as noted above, some propagated replicates gave distinct compositions. Overall, the PCoA revealed an “attractor” composition, and propagation tended to shift the community toward this composition.

The major community members include *Veillonella* OTU002, two *Streptococcus* OTUs (OTU001 and OTU005), *Prevotella* OTU006, and *Alloscardovia* OTU011 (Figure 4). Consistent with the PCoA results, these OTUs had similar relative abundances in Host 2 and Host 3 across samples, generally with <10% of *Veillonella* OTU002, <10% of *Streptococcus* OTU005, and >85% of *Streptococcus* OTU001. In Hosts 2 and 3, no *Alloscardovia* was present above the 0.1% threshold, and there was little or no abundance of *Prevotella* OTU006. In contrast, Host 1 samples contained highly varied, potentially large, abundances of *Streptococcus* OTU001 and *Veillonella* OTU002, with additional smaller contributions from *Streptococcus* OTU005, *Alloscardovia* OTU011, and *Prevotella* OTU006. 

In terms of the effects of preservation and propagation on community composition, propagated cultures from Hosts 2 and 3 almost universally experienced an increase in the relative abundance of the already dominant *Streptococcus* OTU001, as well as a corresponding decrease in the abundance of *Streptococcus* OTU005. In Host 2, propagation also tended to reduce *Veillonella* OTU002, although this OTU was too low in abundance in Host 3 to detect a change. While the culture pellets in Hosts 2 and 3 already had composition resembling the “attractor”, the culture pellets in Host 1 were somewhat different from the “attractor” composition. While preservation did not lead to a systematic change, propagation in Host 1 samples led to large changes favoring *Streptococcus* OTU001 and disfavoring *Veillonella* OTU002. *Prevotella* OTU006 showed some dispersion in response to propagation. Thus, propagation appeared to favor a shift toward a high *Streptococcus*/low *Veillonella* state, although this shift had not reached a steady state in Host 1.

### 3.5. Composition of the “Attractor” Community

To characterize the attractor cluster, principal component analysis (PCA) was performed on the relative abundances of the initial, preserved, and propagated cultures (Figure 5 and Figure 6). Two principal components, analyzed using untransformed relative abundance data, accounted for 99.3% of the variation in the dataset (Appendix A). Principal Component 1 (PC1) in this analysis contained large and opposing contributions from *Streptococcus* OTU001 and *Veillonella* OTU002. Principal Component 2 (PC2) had a major contribution from *Prevotella* OTU006, which opposed smaller contributions from *Streptococcus* OTU001 and *Veillonella* OTU002 (Figure 5). Samples from Hosts 2 and 3 clustered at low values of PC2, while samples from Host 1 varied widely along PC2. 

Since data on relative abundances are internally correlated, we also conducted PCA after applying centered log-ratio (CLR) transformation (Figure 6) or isometric log-ratio (ILR) transformation (Appendix A). CLR transformation resulted in a greater spread of data compared to non-transformed data, with the first two components accounting 57% of the total variation (compared to >99% without transformation). Consistent with the non-transformed data, *Streptococcus* OTU001 and *Veillonella* OTU002 were major contributors of the first component while *Prevotella* OTU006 was still the major factor for the second component (and also an important contributor to the first component). Moreover, as before, *Streptococcus* OTU001 and *Veillonella* OTU002 were found to lie in opposite directions, and both of these OTUs were somewhat orthogonal to *Prevotella* OTU006. These results suggest that a *Streptococcus* taxon may vary inversely with a *Veillonella* taxon, and dominance of the *Prevotella* taxa may displace both *Streptococcus* and *Veillonella* when measuring relative abundance. However, CLR transformation did point out additional contributors to the first and second components. These included *Enterobacteriaceae* OTU004, *Streptococcus* OTU001, and *Alloscardovia* OTU011 as factors contributing to variation across samples.

In the transformed PCA, differences across hosts were also consistent with the non-transformed analysis, although less pronounced. Host 1 samples showed inclination toward the *Veillonella* OTU, as observed in the PCoA and relative abundances, as well as high variation along the direction of the second component, compared to Hosts 2 and 3 (Figure 6a). As with the non-transformed analysis, preservation conditions did not lead to clear clustering (Figure 6b). ILR transformation supported trends observed after CLR transformation (Appendix A). To quantify the influence exerted on community compositions by different hosts or different preservation conditions, we conducted ANOSIM significance tests with Bray–Curtis distance measures. The results indicated that compositional differences were not greatly influenced by preservation condition (correlation coefficients of 0.05505 and 0.1287 for non-rarefied and rarefied relative abundances, respectively). However, consistent with PCoA and PCA, the compositional differences were influenced to a greater extent by host differences (correlation coefficients of 0.3365 and 0.3147 for non-rarefied and rarefied relative abundances, respectively). These results confirmed that host-based variation was a greater source of variation across all samples, compared to preservation condition.

## 4. Discussion

Storage, preservation, and propagation are necessary steps for the creation of experimental microcosms from microbiome samples. Preservation of both natural and artificial human gut microbiota has been studied, including cryopreservation [57,58], lyophilization [57], and long-term storage in commercial storage media [60]. However, despite some efforts [74,75], preservation of the oral microbiome has been relatively understudied, and little is known about the stability of microcosms derived from the oral microbiome. In particular, a method to develop a microcosm of reproducible, convergent composition from plaque samples sourced from different individuals has not been reported yet.

This study reports an initial pilot experiment to probe whether cold preservation and culturing of communities derived from healthy hosts could produce a consistent community composition. A 72 h incubation time (initial cultures) was found to be sufficient to observe transitions in community composition. In particular, the community diversity for all three hosts decreased markedly when comparing the original plaque samples with the initial cultures. The decreased diversity is expected due to the incubation in artificial media, which represents a selective condition that presumably favors some subset of all oral species. The eleven taxa present in the host plaque dropped to five total taxa in the initial cultures, and for all hosts, the five taxa included two *Streptococcus* OTUs and one *Veillonella* OTU. However, the composition of the 72 h culture still depended on the plaque composition, as plaque that contained higher abundances of streptococcal bacteria led to correspondingly high abundances of the *Streptococcus* taxa (Figure 2, Plaque and Cx Pellet, Host 2 and Host 3), whereas plaque with lower proportions of *Streptococcus* but higher proportions of *Veillonella* OTUs yielded initial cultures with much higher abundances of *Veillonella* (Figure 2, Plaque and Cx Pellet, Host 1). Interestingly, two low-abundance taxa in Host 1 plaque were selectively cultivated from the plaque community at this stage, in addition to the *Streptococcus* and *Veillonella* OTUs, namely *Alloscardovia* and *Prevotella*. However, each of these two OTUs appeared in only one replicate of the initial culture (Figure 2, Host 1, Cx Pellet), illustrating the possibility of experimental variation when generating a microcosm.

Preserving the initial cultures did not lead to major membership changes, indicating that the selective pressures from cold preservation did not greatly vary among members that survived the initial culturing process. In contrast, propagation of preserved cultures led to changes in membership and relative abundance. In many samples from all three hosts, propagation resulted in the disappearance of one or more OTUs or a major shift in the relative abundances of OTUs. In Host 2 and 3 samples, *Streptococcus* OTU001 continued to dominate while the abundance of one of the other two OTUs, *Streptococcus* OTU005 and *Veillonella* OTU002, decreased dramatically or became undetectable. In Host 2, propagation of cryopreserved samples led to cultures with higher diversity than propagation of refrigerated samples (Figure 2, Host 2, “Prop”), suggesting that freezing may have preserved greater diversity than refrigeration, although this tendency was not clearly seen for Host 3. In Host 1, cultures propagated from refrigerated samples exhibited shifts generally favoring *Streptococcus* OTU001 and simultaneously disfavoring *Veillonella* OTU002 while the abundance of *Streptococcus* OTU005 remained somewhat stable, and the two additional OTUs (*Prevotella* OTU006 and *Alloscardovia* OTU011) tended to drop in abundance, sometimes to undetectable levels. Like with Host 2, cryopreservation of Host 1 initial cultures, followed by propagation, yielded community compositions that better reflected the initial cultures, and with one exception, propagation did not change the membership of cryopreserved Host 1 samples. Similar to the samples propagated from the refrigerated samples, propagation of the cryopreserved samples led to an increase in the relative abundance of *Streptococcus* OTU001 and a decrease in *Veillonella* OTU002, with one notable exception where a decrease in *Streptococcus* OTU001 was accompanied by an increase in *Prevotella* OTU006 (1.5 Wks, Pellet 2). This change was not observed in other preserved samples of Host 1 where the *Prevotella* taxon was present in substantial proportions. It is possible that organisms from the *Prevotella* genus are less robust under cold preservation, especially refrigeration, or that the propagated cultures needed additional incubation to form a more stable membership. Overall, *Streptococcus* OTUs were the most robust through the preservation and propagation processes, and *Veillonella* OTUs also tended to remain. The *Prevotella* OTU seemed robust to preservation (Figure 2, Host 1, “Pres” compared to “Cx Pellet”), but propagation led to compositions similar to communities lacking this OTU.

A particular community composition, termed the “attractor” community, emerged from Hosts 2 and 3 and from many propagated samples of Host 1. This composition was much simpler than the composition of the original plaque from any host. The attractor community was characterized in particular by two *Streptococcus* OTUs and one *Veillonella* OTU. While the relative abundance shifts in our experiments seem to implicate a trade-off between *Veillonella dispar* (*Veillonella* OTU002) and *Streptococcus salivarius* (*Streptococcus* OTU001), there has been some evidence that biofilms of *S. salivarius* form with greater biomass in the presence of *V. dispar* [76]. Additionally, the other *Streptococcus* OTU in the attractor community, corresponding to *S. oralis* (*Streptococcus* OTU005), has also been shown to co-aggregate with organisms from the *Veillonella* genus [23]. Interestingly, a bacteriocin produced by *S. salivarius*, a commensal organism in the oral cavity [77], is known to target microorganisms with which it does not cooperate [78,79,80], representing a potential selective pressure created by the community itself. Because the organisms were incubated in a complex, mixed community, microbial interactions may also affect the robustness of an organism toward low-temperature, desiccation, or nutrient depletion stresses, including those between *Veillonella* and *Streptococcus* species [81] or between *Streptococcus* and *Actinomyces* species [82]. Thus, selective pressures may arise from the culturing and propagation environment, as well as from direct and indirect interactions among community members. 

Two taxa were observed in propagated Host 1 samples, which were not characteristic of the core attractor community, namely the *Prevotella* and *Alloscardovia* taxa. For communities (plaque or initial cultures) that lacked appreciable abundances of these OTUs, preservation and propagation tended to yield communities consisting primarily of *Streptococcus* and *Veillonella* OTUs. However, for those that contained appreciable *Prevotella* or *Alloscardovia* OTUs, preservation and propagation tended to retain or even increase the abundances of these OTUs. Their persistence is of interest, as both have been linked to oral diseases [83,84]. Furthermore, organisms of the *Prevotella* genus may be dependent on other organisms, such *Fusobacterium* [26], which also co-aggregate with *Veillonella* [22] and are implicated in oral diseases. Indeed, other studies have shown that normal human oral microflora is dominated by *P. melaninogenica*, *P. histicola*, and *P. intermedia*, but *P. salivae* [85], the *Prevotella* species observed in Host 1 samples, is not a dominant member of normal microflora. The ability of these taxa to co-exist, even transiently, with *Streptococcus* and *Veillonella* suggests that pathogenic or opportunistically pathogenic taxa may be culturable in a microcosm. 

In the bioinformatic analysis, we applied rarefaction due to the observation of spurious OTUs in the known mock communities at higher read counts. Although rarefaction may not always be appropriate [69], the inclusion of both negative and positive controls here enabled a rarefaction application to accurately represent the number of OTUs in the samples. Since the “attractor” community composition was observed to have a small number of OTUs, similar to the mock community, rarefaction to a small number of reads could still capture the community composition. Given the proliferation of “micro” high-throughput sequencing services, the rarefaction employed here indicates that similar studies could be performed at low cost for resource-limited environments. Here, we used the mothur pipeline with a naïve Bayesian classifier [33] to assign reads into bins based on read distances with similarity cutoffs [67,86,87,88]. We then matched the OTUs to reference taxa in the complete SILVA database (version 132) [66]. Ordination using inter-sample distances and variance is frequently used to characterize differences among microbiome samples, such as across different body sites and different hosts [89,90,91] or when comparing healthy and diseased states [92,93]. Here, we used principal coordinate analysis [94] and principal component analysis [95] to examine general patterns in sample similarities [96,97,98] and visualize the underlying factors or “components” [99,100]. Both techniques reduced the high dimensionality from the large number of OTUs to a small number of dimensions [101], which captured a majority of the variation in our samples. However, because relative abundance data exist in simplex space where measurements in each sample sum to a constant, these internally correlated data can yield spurious results [102]. This problem can be mitigated by transformation to a Euclidean space with absolute counts, using centered-log-ratio and isometric-log-ratio transformations. Alternatively, techniques such as ANOSIM (analysis of similarities) do not assume a distribution shape for testing the null hypothesis (e.g., no difference between groups) [95]. In the analysis here, applying these techniques generally supported the conclusions drawn from the non-transformed data.

While this small study advances a proof of concept for generating a reproducible microcosm from the oral microbiome, several caveats apply to the experiments reported here. Ideally, the dependence of the incubation time for both the initial and the propagated cultures should be studied in depth to determine the rate of approach to, and composition of, the steady-state community (and the stability of the community). In addition, the inoculum size could be varied to determine whether there exist frequency-dependent effects on the community composition. Indeed, variability among different replicates indicated some source of noise, perhaps in the initial inoculum. Another source of variability could be in the DNA extraction process, which varies in efficiency for different microorganisms. For the experiments here, the cultures were dominated by Gram-positive bacteria, and Gram-positive organisms from the microbial standard were reasonably equitably detected from the mock community. However, experiments using diseased dental plaque may contain a mixture of Gram-positive and Gram-negative organisms and thus may require additional extraction to minimize the tendency toward over-representation of Gram-negative bacteria, such as those associated with dental caries and periodontal diseases (e.g., *Porphyromonas*, *Trepomona*, *Tannerella,* and *Lactobacillus* taxa [103,104]). Absolute abundance, rather than relative abundance, would be preferable for analysis to avoid spurious correlations among OTUs and might be achieved using internal sequencing standards and/or cell counting. Finally, the composition of the “attractor” community emerging in this study should be confirmed across additional individuals and greater culturing times.

## 5. Conclusions

The establishment of microcosm models of the microbiome is an important goal for creating useful, yet experimentally tractable, microbial communities to understand systems’ properties and interactions. For microcosms created by seeding from a complex and variable community, incubation, cold storage and preservation, and propagation all represent potentially selective conditions that influence the resulting microcosm. Characterizing how these steps affect the composition and variability of a microcosm is important for establishing experimental models. The data reported here may serve as a starting point for developing methods to create reproducible microcosms from variable inocula.

## Figures and Tables

**Figure 1 microorganisms-10-02146-f001:**
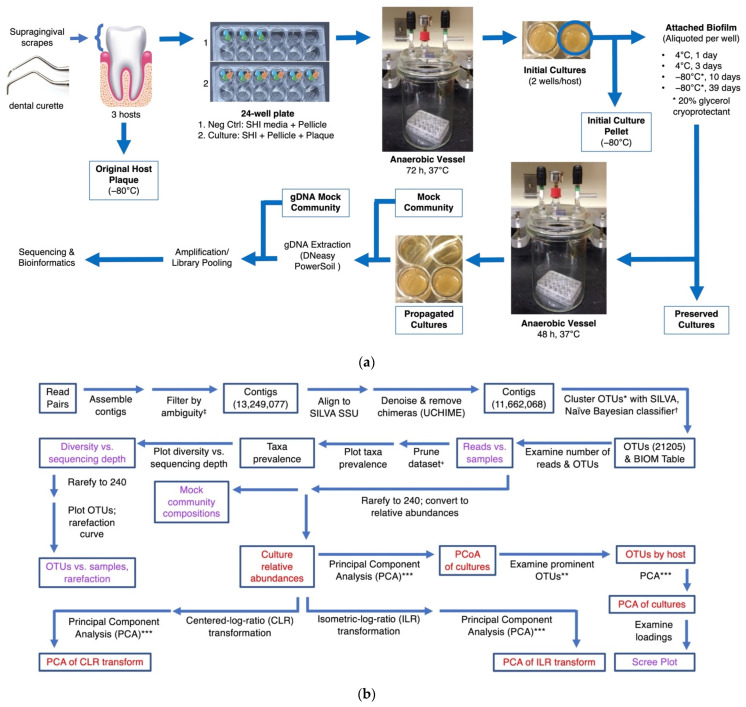
Experimental design and bioinformatic analysis. (**a**) Experimental scheme for preservation and propagation experiments; blue boxes around text indicate samples that were analyzed. Negative controls and host cultures were grown in separate plates to minimize cross-contamination, and samples not used for downstream culturing were flash-frozen in 20% glycerol and stored at −80 °C until further processing. Attached and/or sedimented cells in each well were divided into five aliquots, four of which underwent a preservation type and one that was reserved for later analysis. (**b**) Bioinformatics pipeline and statistical analyses. The SILVA SSU referenced here is the non-redundant database (version 132). Purple: figures with quality-control results; red: figures included in the final results of bioinformatics analysis. * 3% dissimilarity; ** analysis only includes OTUs with relative abundances greater than 0.1%; *** analysis includes OTUs with relative abundances greater than 0.1% (after rarefaction), cultures only. ^†^ 80% pseudo-bootstrap; ^‡^ reads with ambiguously assigned bases are filtered out at this step. ^+^ Prune out all controls and plaque samples, samples with <1000 reads, and one outlier (more than 700,000 reads).

**Figure 2 microorganisms-10-02146-f002:**
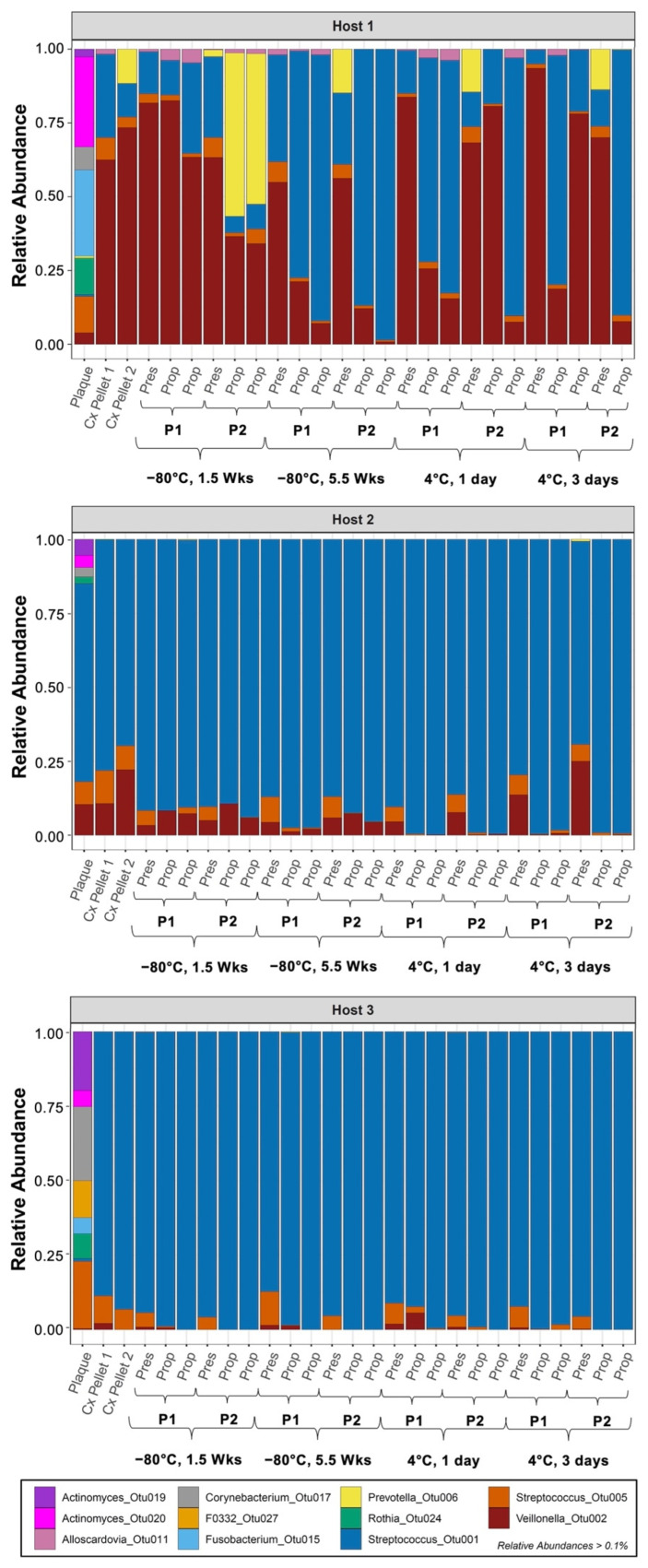
Microbial community composition of plaque and cultures, for OTUs with relative abundances greater than 0.1% (after rarefaction). The bars of each subfigure (for Hosts 1–3) show the abundance of the plaque sample used to inoculate the initial culture, initial cultures from two replicate wells (Cx Pellet 1 and Cx Pellet 2), and multiple preservation and propagation conditions. “Cx” = initial culture with 72 h incubation; “Pres” = preserved culture; “Prop” = propagated culture with 48 h incubation; P1/P2 = Pellet 1/Pellet 2, from two adjacent wells in the same plate serving as duplicates for each preservation condition. Each of the two wells from preservation conditions was used to inoculate two duplicate wells for propagation, ultimately leading to two “preserved” and four “propagated” samples for each preservation condition. One duplicate propagated culture sample from Host 1 (4 °C, 3 days, Pellet 2) was excluded from analysis due to insufficient number of reads (<240). Host 2 and 3 cultures were dominated by *Streptococcus* OTUs and, in some cases, one *Veillonella* OTU, while Host 1 cultures generally contained higher abundances of *Veillonella*, as well as presence of *Prevotella* and Alloscardovia.

**Figure 3 microorganisms-10-02146-f003:**
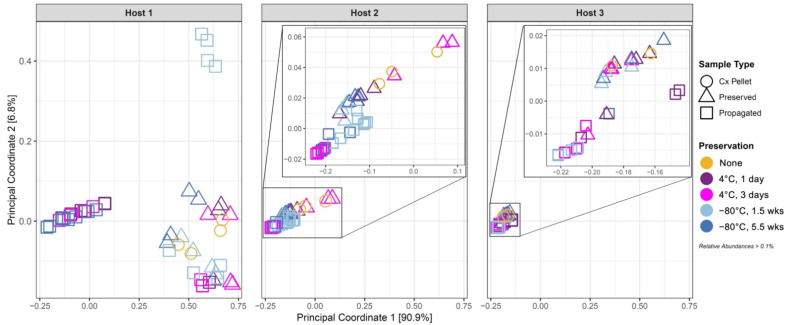
Principal coordinate analysis (PCoA) using Bray–Curtis distance measurement, performed on all samples as a single dataset.

**Figure 4 microorganisms-10-02146-f004:**
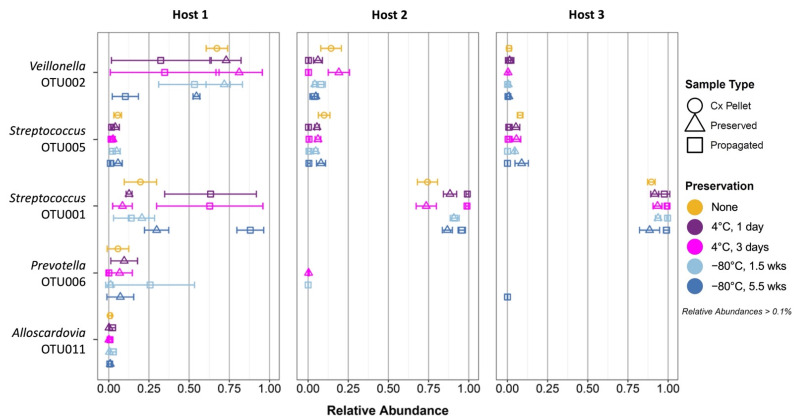
Relative abundances of the most prominent OTUs for each host. “Cx Pellet”: initial culture. Averages +/− standard deviations are shown by horizontal bars.

**Figure 5 microorganisms-10-02146-f005:**
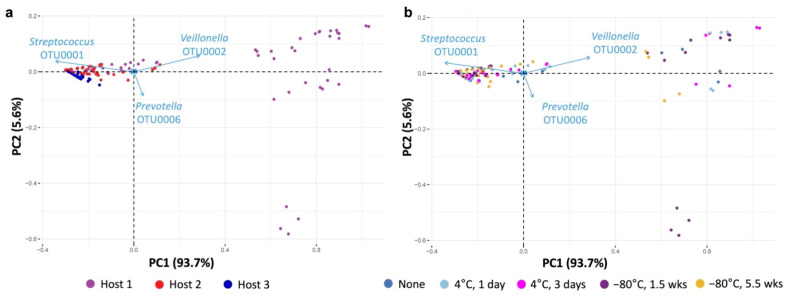
PCA biplots of relative abundances, by host (**a**) or preservation condition (**b**). Each dot represents a single sample (see legend). Arrows depict major OTUs in the dataset. Note the presence of a cluster of diminutively short arrows near the origin, containing OTUs that do not contribute substantially to the variation in the dataset. Dimension 1: *Streptococcus* OTU001 (–), *Veillonella* OTU002 (+); Dimension 2: *Prevotella* OTU006 (–), *Veillonella* OTU002 (+), *Streptococcus* OTU001 (+).

**Figure 6 microorganisms-10-02146-f006:**
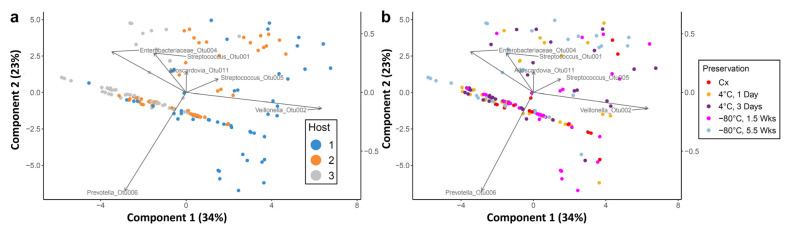
PCA of relative abundances after centered-log-ratio transformation, by (**a**) host or (**b**) preservation condition. Major patterns are similar to the non-transformed analysis, but more contributing factors were observed, including the *Alloscardovia* OTU and an *Enterobacteriaceae* OTU.

## Data Availability

All data generated in the study and code used to perform the analysis are freely available at the Dryad Digital repository at the following link: https://doi.org/10.25349/D9P02H.

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
