# Peer review of "Effects of Preservation and Propagation Methodology on Microcosms Derived from the Oral Microbiome"

_microorganisms, 2022, doi:10.3390/microorganisms10112146_

Round 1

Reviewer 1 Report

The manuscript describes the in vitro (supragingival dental plaque) microcosms derived from 3 participants to investigate the reproducibility of a microcosm derived from preserved, complex bacterial communities. The number of participants seems rather small, but the manuscript is well written (in detail enough) and suitable for publication in the Microorganisms.

[Suggestions]
L. 98-101:
Is it possible for the authors to describe the age and gender of the "three healthy adult hosts"?

Typographical errors
L. 187:
"16. S rRNA Bioinformatics Analysis" should read "16S rRNA ...".

Author Response

We thank the reviewer for the helpful evaluation. The revised manuscript is attached to this response.

  1. 98-101:
    Is it possible for the authors to describe the age and gender of the "three healthy adult hosts"?

The recruitment criteria were adults age 18-80, and gender was not specified. We added this information in the Methods section. Data on specific ages and genders were not collected.

Typographical errors
L. 187:
"16. S rRNA Bioinformatics Analysis" should read "16S rRNA ...".

This appears correctly in the Word document manuscript, so it is likely to be a problem with the conversion of the submitted Word document into the journal’s PDF format. We will make sure to check for this error if the manuscript is accepted into the proof stage.

Reviewer 2 Report

This paper is very well written and incredibly interesting. I commend the authors on their solid work. A few small things.

Abstract- the word 'attractor' is used but not clarified

Figure 1b. Can this separated into another figure and be made more linear? I like 1a but 1b is very tough to decipher. 

I could not identify any other methodological or interpretation issues. 

Nicely done.

Author Response

We thank the reviewer for the helpful evaluation. The revised manuscript is attached to this response.

Abstract- the word 'attractor' is used but not clarified

We have reworded this section to introduce the word ‘attractor’.

Figure 1b. Can this separated into another figure and be made more linear? I like 1a but 1b is very tough to decipher. 

We have increased the white space between Figure 1A and 1B, and we revised Figure 1B into a horizontal format to increase the linearity. We adjusted the fonts to improve readability as well.
